biomathematics

DENV, ZIKV, mathematical model, antibody-dependent enhancement, viral dynamics, parameter estimation

**Author for correspondence:**
Jianhong Wu
e-mail: wujh@yorku.ca

# Modelling the impact of antibody-dependent enhancement on disease severity of Zika virus and dengue virus sequential and co-infection

Biao Tang[1,2,3], Yanni Xiao[4], Beate Sander[2,3], Manisha A. Kulkarni[5], RADAM-LAC Research Team and Jianhong Wu[1]

[1]Laboratory for Industrial and Applied Mathematics (LIAM), York University, Toronto, Canada
[2]Institute of Health Policy, Management and Evaluation, University of Toronto, Toronto, Canada
[3]Toronto Health Economics and Technology Assessment, Toronto, Ontario, Canada
[4]School of Mathematics and Statistics, Xi'an Jiaotong University, Xi'an, 710049, People's Republic of China
[5]School of Epidemiology and Public Health, University of Ottawa, Ottawa, Canada

YX, 0000-0003-0432-7628; JW, 0000-0003-0052-5336

Human infections with viruses of the genus *Flavivirus*, including dengue virus (DENV) and Zika virus (ZIKV), are of increasing global importance. Owing to antibody-dependent enhancement (ADE), secondary infection with one *Flavivirus* following primary infection with another *Flavivirus* can result in a significantly larger peak viral load with a much higher risk of severe disease. Although several mathematical models have been developed to quantify the virus dynamics in the primary and secondary infections of DENV, little progress has been made regarding secondary infection of DENV after a primary infection of ZIKV, or DENV-ZIKV co-infection. Here, we address this critical gap by developing compartmental models of virus dynamics. We first fitted the models to published data on dengue viral loads of the primary and secondary infections with the observation that the primary infection reaches its peak much more gradually than the secondary infection. We then quantitatively show that ADE is the key factor determining a sharp increase/decrease of viral load near the peak time in the secondary infection. In comparison, our simulations of DENV and ZIKV co-infection (simultaneous rather than sequential)

show that ADE has very limited influence on the peak DENV viral load. This indicates pre-existing immunity to ZIKV is the determinant of a high level of ADE effect. Our numerical simulations show that (i) in the absence of ADE effect, a subsequent co-infection is beneficial to the second virus; and (ii) if ADE is feasible, then a subsequent co-infection can induce greater damage to the host with a higher peak viral load and a much earlier peak time for the second virus, and for the second peak for the first virus.

## 1. Introduction

Dengue virus (DENV), transmitted by *Aedes aegypti* and *Aedes albopictus* mosquitoes, infects 50–100 million people yearly, including 500 000 dengue hemorrhagic fever (DHF) cases and 22 000 deaths [1,2]. Zika virus (ZIKV), also a member of the Flaviviridae family, genus *Flavivirus*, is transmitted by the same mosquitoes. Human infection with ZIKV is usually accompanied by relatively mild symptoms, but can be associated with more severe effects such as Guillain–Barré syndrome and fetal microcephaly [3,4], prompting global concern.

Infection with any of the four closely related dengue serotypes (DENV 1, DENV 2, DENV 3 and DENV 4) induces protective immunity to that serotype, but confers no long-term protection against infection by other serotypes. Experimental evidence [5] has indicated that infection with one dengue serotype provides a temporal window of cross-protection towards other dengue serotypes, and the study of Reich *et al.* [6] provided the quantitative measure of short-term cross-protection among the different dengue serotypes.

By contrast, several studies [7–9] have reported that prior ZIKV infection can induce significant low levels or no cross-neutralizing effect of secondary infection with any dengue serotype, suggesting that ZIKV lies outside the DENV serocomplex [8]. In [10], Dejnirattisai *et al.* concluded that most antibodies which reacted to the DENV envelope protein also reacted to ZIKV. More specifically, DENV-specific antibodies can bind ZIKV but are unable to neutralize the virus, and consequently facilitate ZIKV infection with a high level of Zika viral loads in the host. This phenomenon is referred as antibody-dependent enhancement (ADE) [10–13]. Correspondingly, Valiant *et al.* [9] showed that ZIKV-exposed macaques present a high level of DENV cross-reactive binding antibody with low DENV neutralizing activity, indicating the occurrence of enhancement of the dengue infection. In addition, George *et al.* [14] showed that prior exposure to ZIKV significantly enhances DENV viremia. It is accepted that ADE has been well-documented among different dengue serotypes [15–17]. In particular, driven by ADE, a secondary infection of dengue with a different serotype from the first infection is much more severe than the primary infection, and has been linked with the increase in DHF [18,19].

On the one hand, several mathematical epidemiological modelling studies [20–23] have examined the epidemiological impact of ADE on the prevalence and persistence of different dengue serotypes at the population level. In the studies [20,21], the authors showed that ADE can induce large-amplitude oscillations and other complex long-term behaviours in the incidence. By extending the model in [21], Billings *et al.* were able to conduct some computational analyses to suggest optimal vaccination strategies [23]. Also, a modelling study reported that a dengue vaccine used in a population may increase ZIKV outbreaks under certain conditions owing to ADE [24]; however, the work [25] also showed that an appropriately designed and optimized dengue vaccination programme can not only help control the dengue spread but also reduce ZIKV infections. On the other hand, several within-host mathematical models were proposed and used to quantitatively analyse [26–28] or theoretically investigate [29–31] the impact of ADE on the viral dynamics of primary and secondary infection of different dengue serotypes. Note that, both studies [28,31] highlighted the important role of antibody in controlling the viral replication. However, to the best of our knowledge, there is no mathematical study evaluating the impact of ADE on the viral dynamics in the secondary infection or the co-infection of DENV and ZIKV. This study aims to quantitatively address these issues.

The rest of the paper is organized as follows. In the coming section, we propose models describing virus dynamics for a primary infection of DENV; for a secondary infection of DENV with a previous infection of ZIKV; and for co-infection of DENV and ZIKV. In §3, we calibrate our models by fitting them to some data of dengue viral loads. In §4, through a sensitivity analysis (SA) and some numerical simulations, we discuss the impact of ADE on the peak value and time of dengue viral loads. Finally, in §5, we summarize the main results and elaborate these modelling analyses in the context of viral dynamics.

# 2. Model formulation

## 2.1. Primary infection of single virus

The within host viral dynamics of *Flaviviruses*, including dengue [26–28] and Zika [32,33], has been studied intensively. The viral dynamics with antibody mediated immune response is described by the following ordinary differential equation system:

$$
\left.
\begin{aligned}
\frac{dT(t)}{dt} &= \Lambda - \mu T(t) - \beta V(t)T(t), \\
\frac{dI(t)}{dt} &= \beta V(t)T(t) - \delta I(t), \\
\frac{dV(t)}{dt} &= \omega I(t) - cV(t) - bA(t)V(t) \\
\frac{dA(t)}{dt} &= aV(t)A(t) - \sigma A(t),
\end{aligned}
\right\}
\tag{2.1}
$$

and

where $T$ is the target cell, $V$ the free virus, $I$ the infected cells and $A$ the antibody specific to the virus $V$. Here, we assume that the target cells are recruited at a constant rate $\Lambda$ and die at the rate $\mu$, free viruses infect the target cells through a mass-action progress at a rate $\beta$, $\delta$ denotes the death rate of the infected cell, $\omega$ is the production rate of the free virus, and $c$ is the clearance rate of free viruses. The antibody $A$ can be stimulated by the free virus with a production rate $a$ while it declines at a rate of $\sigma$. The antibody $A$ can neutralize the free virus at the rate of $b$.

## 2.2. Secondary infection with antibody-dependent enhancement

We focus on the case of a secondary DENV infection with a primary ZIKV infection. Without loss of generality, we assume that the individual has recovered from ZIKV, hence, free ZIKV has been cleared. After the secondary infection with DENV, antibodies specific to it will be activated. Meanwhile, owing to the cross-immune response between DENV and ZIKV, antibodies specific to ZIKV will be stimulated by the free DENV as well. As mentioned in the introduction, ZIKV-specific antibodies can bind DENV, and consequently, contribute to the replication of DENV (i.e ADE effect) [9]. On the other hand, as demonstrated in the experimental study [9], the delayed cross-neutralization of ZIKV-specific antibody to DENV occurs in the secondary DENV infection. Hence, it is reasonable to assume that at a high level of ZIKV-specific antibody, it can help in clearing the DENV (antibody-dependent neutralization (ADN)). As a conclusion, we assume that the ZIKV-specific antibody can present ADE to DENV only when its concentration reaches a certain level. Once the concentration of ZIKV-specific antibody exceeds a threshold level, it can help in neutralizing DENV (ADN). Therefore, based on the above assumptions, we propose the follow model describing the within-host dynamics of the secondary DENV infection with a primary infection of ZIKV:

$$
\left.
\begin{aligned}
\frac{dT(t)}{dt} &= \Lambda - \mu T(t) - \beta_d V_d(t)T(t), \\
\frac{dI_d(t)}{dt} &= \beta_d V_d(t)T(t) - \delta I_d(t), \\
\frac{dV_d(t)}{dt} &= \omega_d I_d(t) - c_d V_d(t) - b_d A_d(t)V_d(t) + \theta_d S A_z(t)V_d(t), \\
\frac{dA_d(t)}{dt} &= a_d V_d(t)A_d(t) - \sigma_d A_d(t) \\
\frac{dA_z(t)}{dt} &= \kappa V_d(t)A_z(t) - \sigma_{sz} A_z(t),
\end{aligned}
\right\}
\tag{2.2}
$$

and

where $T$ is the target cells, $I_d$ the cells infected by DENV, $V_d$ the free DENV, $A_d$ the DENV-specific antibody and $A_z$ the ZIKV-specific antibody. A schematic diagram of the secondary infection of DENV with a primary infection of ZIKV is shown in figure 1$a$. Here, $\kappa$ is the production rate of ZIKV-specific antibody $A_z$ owing to its cross-immune response to DENV and $\sigma_{sz}$ denotes the clearance rate of ZIKV-specific antibody in the secondary infection of DENV. $\theta_d$ denotes the maximal changing

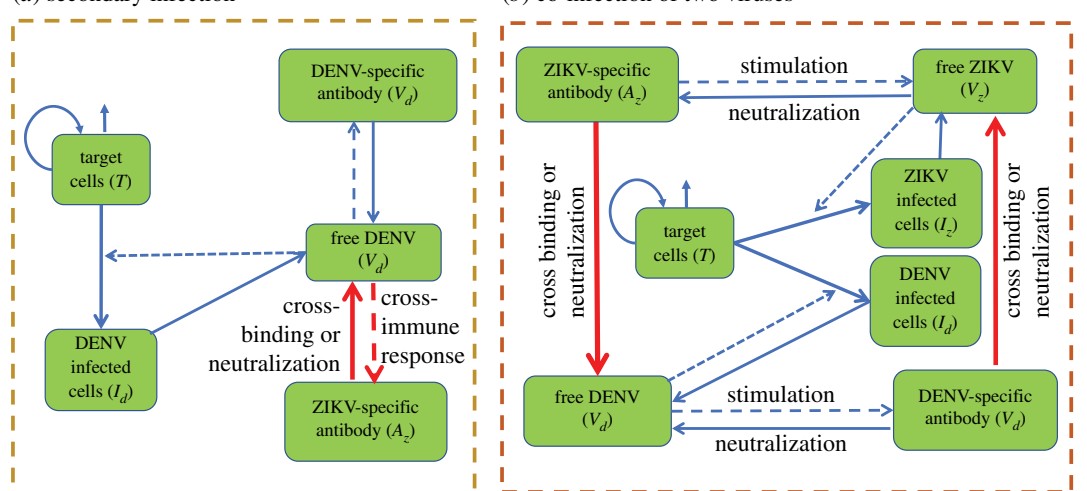

**Figure 1.** Schematic diagrams. (*a*) Schematic diagram for the secondary infection of DENV with a primary infection of ZIKV. (*b*) Schematic diagram of the co-infection of DENV and ZIKV.

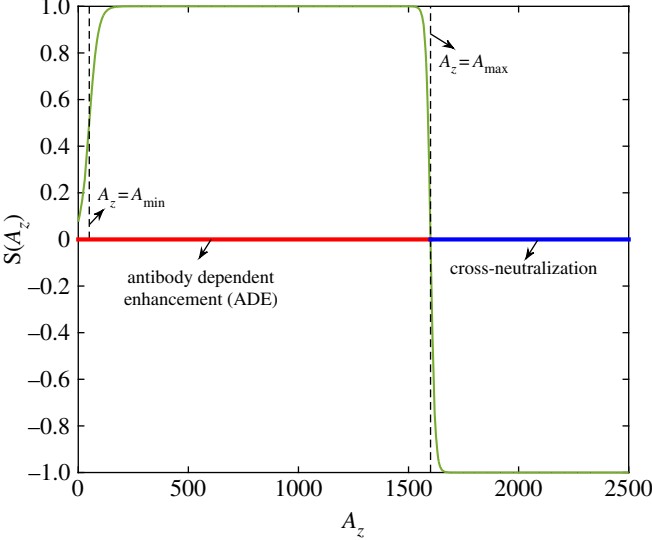

**Figure 2.** Shape of the switching function $S(A_z)$. Here, we fix $A_{min} = 50$ and $A_{max} = 1600$.

rate of free DENV owing to the cross-reactive response of ZIKV-specific antibody to DENV. $S$ is a switching function of $A_z$ with the following form [29]:

$$S(A_z) = \frac{-\tanh(\alpha_1(A_z - A_{max}))}{1 + e^{-\alpha_2(A_z - A_{min})}}. \tag{2.3}$$

It is used here to describe the continuous switching between the ADE and the ADN of ZIKV-specific antibody to DENV. In this study, we fixed $\alpha_1 = \alpha_2 = 0.05$ so that as $A_z$ increases, $S(A_z)$ first increases when $A_z$ is less than $A_{min}$, reaches and remains at the value of 1 for a certain interval of $A_z$, then decreases to 0 when $A_z$ touches $A_{max}$, and finally decrease to $-1$, as shown in figure 2. Correspondingly, $\theta_d S$ denotes the replication rate of free DENV owing to the ADE of ZIKV-specific antibody to DENV when $A_z \in (0, A_{max})$, while it represents the clearance rate of free DENV owing to the cross-neutralization of ZIKV-specific antibody to DENV when $A_z > A_{max}$. Thus, $A_{max}$ is the threshold level of ZIKV-specific antibody for presenting ADE or ADN to DENV. Note that, for the very low concentration of the heterologous antibodies, the probability of its interaction with virus should be low or zero. $A_{min}$ can be seen as a modification parameter for the low concentration of ZIKV-specific antibody presenting low level of ADE. The infection dynamics of DENV and the DENV-specific immune response are treated to be similar to the primary DENV infection, hence the definitions of other parameters are similar to those in model (2.1), which are listed in table 1.

**Table 1.** Parameters definitions and values.

| | definitions | mean (std) | references |
|---|---|---|---|
| **parameters** | | | |
| $\Lambda$ | the rate at which the target cells are created | 57865 (4304) | estimated by model (2.1) |
| $\mu$ | the death rate of the target cells | 0.14 | [28] |
| $\delta$ | the death rate of infected cells | 0.14 | [28] |
| $\beta_d$ | the infection rate of DENV | $2.34 \times 10^{-9}$ ($1.29 \times 10^{-10}$) | estimated by model (2.1) |
| $\beta_z$ | the infection rate of ZIKV | $\beta_d$ | assumed |
| $\omega_d$ | production rate of DENV | $1 \times 10^4$ | [28] |
| $\omega_z$ | production rate of ZIKV | $1 \times 10^4$ | [28] |
| $c_d$ | clearance rate of DENV | 10 | [32,34] |
| $c_z$ | clearance rate of ZIKV | 10 | [32,34] |
| $\sigma_d$ | decay rate of DENV-specific antibody $A_d$ | $3.99 \times 10^{-4}$ ($2.87 \times 10^{-5}$) | estimated by model (2.1) |
| $\sigma_z$ | decay rate of ZIKV-specific antibody $A_z$ | $\sigma_d$ | assumed |
| $a_d$ | production rate of DENV-specific antibody $A_d$ | $5.56 \times 10^{-5}$ ($1.4 \times 10^{-6}$) | estimated by model (2.1) |
| $a_z$ | production rate of ZIKV-specific antibody $A_z$ | $a_d$ | assumed |
| $b_d$ | neutralization rate of DENV-specific antibody to DENV | 0.95 (0.1) | estimated by model (2.1) |
| $b_z$ | neutralization rate of ZIKV-specific antibody to ZIKV | $b_d$ | assumed |
| $\theta_d$ | maximum changing rate of DENV owing to cross-reactive response of $A_z$ to it | 0.343 (0.078) | estimated by model (2.2) |
| $\theta_z$ | maximum changing rate of ZIKV owing to cross-reactive response of $A_d$ to it | $\theta_d$ | assumed |
| $\kappa$ | production rate of ZIKV-specific antibody owing to cross-immune response | $7.86 \times 10^{-5}$ ($1.01 \times 10^{-5}$) | estimated by model (2.2) |
| $\sigma_{sz}$ | clearance rate of ZIKV-specific antibody in the second infection of DENV | $5.5 \times 10^{-4}$ ($1.9 \times 10^{-5}$) | estimated by model (2.2) |
| $A_{min}$ | parameter for low ZIKV-specific antibody presenting low level of ADE | 35.3 (2.83) | estimated by model (2.2) |
| $A_{max}$ | threshold value between ADE and ADN of ZIKV-specific antibody to DENV | 1580.2 (143.57) | estimated by model (2.2) |
| **initial values** | | | |
| $T(0)$ | initial DENV-infected cells | $1.98 \times 10^6$ ($1.14 \times 10^5$) | estimated by model (2.1) |
| $I_d(0)$ | initial DENV-infected cells | 0 | assumed |
| $V_d(0)$ | initial density of DENV | 1 | assumed |
| $A_d(0)$ | initial DENV-specifical antibody | 0.988 (0.056) | estimated by model (2.1) |
| $A_z(0)$ | initial ZIKV-specifical antibody (for model (2.2) only) | 4.83 (0.79) | estimated by model (2.2) |

## 2.3. Co-infection of dengue virus and Zika virus with antibody-dependent enhancement

Owing to the short-term cross-protection of different dengue serotypes, it is considered to be rare that one individual is infected with two or more different dengue serotypes simultaneously. However, differing from the relationship among different DENV serotypes, the no or low level cross-protection between DENV and ZIKV indicates the possibility of the co-infection of these two viruses [9,35,36]. The virus dynamics of the co-infection of DENV and ZIKV can be described as follows:

$$
\left.\begin{aligned}
\frac{dT(t)}{dt} &= \Lambda - \mu T(t) - \beta_d V_d(t)T(t) - \beta_z V_z(t)T(t), \\
\frac{dI_d(t)}{dt} &= \beta_d V_d(t)T(t) - \delta I_d(t), \\
\frac{dI_z(t)}{dt} &= \beta_z V_z(t)T(t) - \delta I_z(t), \\
\frac{dV_d(t)}{dt} &= \omega_d I_d(t) - c_d V_d(t) - b_d A_d(t)V_d(t) + \theta_d S(A_z)A_z(t)V_d(t), \\
\frac{dV_z(t)}{dt} &= \omega_z I_z(t) - c_z V_z(t) - b_z A_z(t)V_z(t) + \theta_z S(A_d)A_d(t)V_z(t), \\
\frac{dA_d(t)}{dt} &= a_d V_d(t)A_d(t) - \sigma_d A_d(t) \\
\frac{dA_z(t)}{dt} &= a_z V_z(t)A_z(t) - \sigma_z A_z(t),
\end{aligned}\right\}
\tag{2.4}
$$

and

where $T$, $V_d$, $V_z$, $I_d$, $I_z$, $A_d$, $A_z$ denote the target cells, free DENV, free ZIKV, DENV-infected cells, ZIKV-infected cells, DENV-specific antibody and ZIKV-specific antibody, respectively. A schematic diagram of the co-infection of DENV and ZIKV is shown in figure 1$b$. Here, we take the DENV-specific immune response and the ZIKV-specific immune response to be similar to the primary infection, as the definitions of all the parameters are listed in table 1. Therefore, the production rate and the clearance rate of the ZIKV-specific antibody $A_z$ are different from those in the secondary infection. In addition, we take into consideration the cross-binding effect of the antibodies to the viruses. Here, similarly to the secondary infection, we use the parameter $\theta_d$ and the switching function $S(A_z)$ to describe the ADE effect or ADN effect of ZIKV-specific antibody to DENV. The definition of the parameter $\theta_z$ is similar to the definition of $\theta_d$, while the switching function $S(A_d)$ is of the form in equation (2.3).

# 3. Material and methods

## 3.1. Data

We obtained data on dengue viral loads from an experimental study on macaques infected by DENV [9], as shown in figure 3. There are two columns of dengue viral loads in table 1 reflecting the results of the experimental study [9]. One of these is obtained by testing the macaque infected with DENV only, as shown in figure 3$a$, while the other column is obtained by testing the ZIKV convalescence macaque super-infected with DENV, as shown in figure 3$b$. Comparing the two columns of dengue viral loads, we can easily see that the peak viral load of the secondary infection is significantly higher than the peak viral load of the primary infection.

## 3.2. Parameter estimation procedure

To calibrate the parameters for the primary infection of DENV, that is, all the parameter values of model (2.1) corresponding to DENV infection, we fix the lifespan of target cells and infected cells as 7 days, hence, $\mu = \delta = 1/7 \approx 0.14$ [28]. One infected cell can produce 10 000 virus per day, thus, we set $\omega_d = 1 \times 10^4$ [28,32]. Referring to the modelling study on the within-host dynamics of ZIKV [32] and the study on the within-host dynamics of influenza [34], we fix the clearance rate of DENV as 10 (i.e. $c_d = 10$). At the initial time, we assume that no target cell is infected, thus $I_d(0) = 0$. Furthermore, the initial dengue viral load is arbitrarily set to 1. Hence, the main task is to estimate the remaining parameters of model (2.1), including the recruitment rate $\Lambda$, the infection rate $\beta_d$, the production rate of DENV-specific antibody $a_d$, the neutralization rate of DENV-specific antibody to DENV $b_d$, the decay rate of DENV-specific antibody $\sigma_d$, the initial value of target cells $T(0)$, and the initial DENV-specific antibody $A_d(0)$.

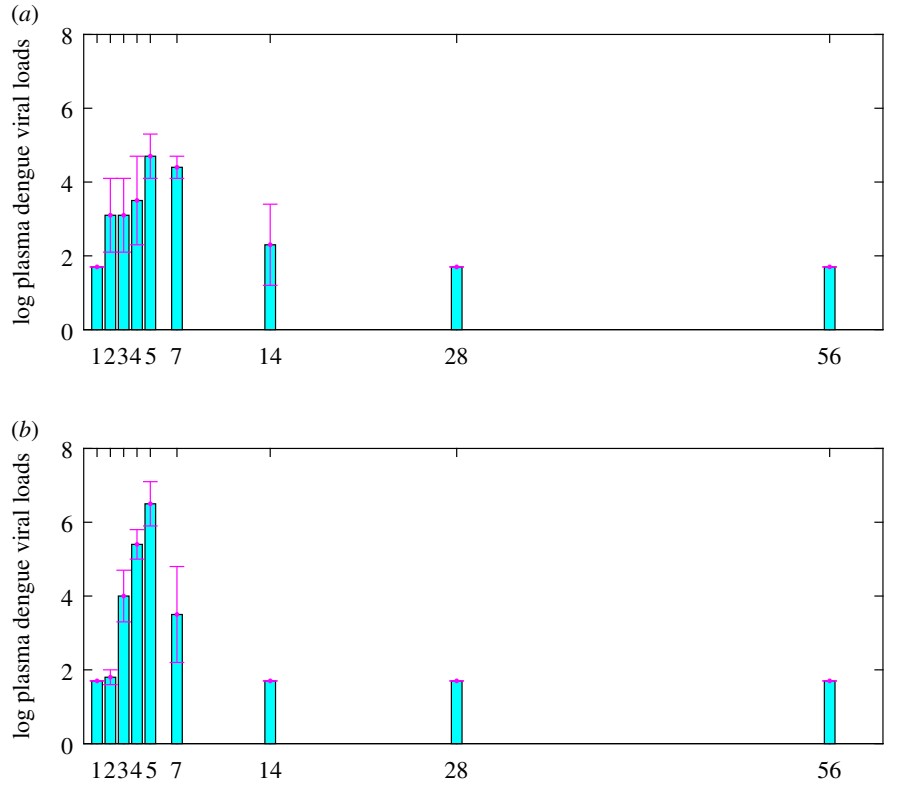

**Figure 3.** Data information from the existing experimental study [9]. (*a*) Data of dengue viral loads from macaque infected with DENV only. (*b*) Data of dengue viral loads from ZIKV convalescence macaque super-infected with DENV. Here, the bars are the mean values of the viral loads on $\log_{10}$-scale while the error bars represent the corresponding standard deviations.

Furthermore, we calibrate the parameters for the secondary infection of DENV with previous infection of ZIKV (i.e. the parameters of model (2.2)), in particular, estimating the parameters corresponding to the cross-immune response. To do this, the parameters related to DENV-infection and DENV-specific immune response are treated in the same manner as those in the primary infection of DENV, that is, all the parameters in model (2.1). This includes $\Lambda$, $\mu$, $\delta$, $\beta_d$, $\omega_d$, $a_d$, $b_d$, $c_d$, $\sigma_d$. Similarly, we fix the initial values of the infected cells as 0 and assume that the initial dengue viral load is 1. We assume that the initial target cell and the initial DENV-specific antibody are of the same values estimated by model (2.1). Therefore, we only need to estimate the production rate of ZIKV-specific antibody owing to cross-immune response $\kappa$, the clearance rate of ZIKV-specific antibody $\sigma_{sz}$, the maximum changing rate of free DENV owing to the cross-reactive response of ZIKV-specific antibody $\theta_d$, the parameters of the switching function $A_{\min}$ and $A_{\max}$, and the initial ZIKV-specific antibody $A_z(0)$.

We use the Markov chain Monte Carlo (MCMC) method to fit the model, and adopt an adaptive Metropolis–Hastings algorithm to carry out the MCMC procedure [37]. The algorithm is run for 800 000 iterations with a burn-in of the first 400 000 iterations, and the Geweke convergence diagnostic method is employed to assess convergence of chains. More specifically, using the MCMC method, we fit model (2.1) to the data of dengue viral load from macaque infected with DENV only, as shown figure 4*a*. We then estimate the unknown parameter values of model (2.1) and their standard deviations, as listed in table 1. We next fit model (2.2) to the data of dengue viral load from ZIKV convalescence macaque super-infected with DENV, estimate the cross-immune response related parameter values ($\kappa$, $\sigma_{sz}$, $\theta_d$, $A_{\min}$, $A_{\max}$), the initial ZIKV-specific antibody ($A_z(0)$), and their standard deviations, which are listed in table 1 as well.

In addition, we use the coefficient of determination ($R^2$) measure to estimate the goodness of fit for our model fitting results. Given an observed data $y$ (in the $n$-dimensional Euclidean space) and the corresponding estimated values from the model $\hat{y}$, the coefficient of determination ($R^2$) value can be calculated as

$$R^2 = 1 - \frac{SS_{\text{err}}}{SS_{\text{tot}}},$$ 

(3.1)

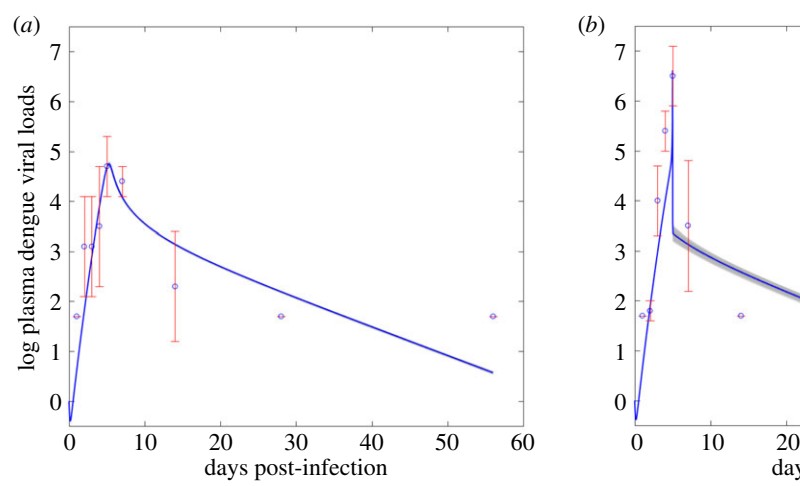

**Figure 4.** Results of model fitting. (*a*) Fitting model (2.1) to the dengue viral loads from macaque infected with DENV only. (*b*) Fitting model (2.2) to the dengue viral loads from ZIKV convalescence macaque super-infected with DENV. The blue circles represent the mean dengue viral loads on log $_{10}$-scale, the error bars are their standard derivations, and the blue curve is the fitting curve.

where $SS_{err}$ and $SS_{tot}$ are the sum of squares of residuals (residual sum of squares) and the total sum of squares (proportional to the sample variance), respectively, which are given by

$$SS_{err} = \sum_i ((y_i - \hat{y}_i)^2) \quad \text{and} \quad SS_{tot} = \sum_i ((y_i - \bar{y})^2),$$

with $\bar{y} = \frac{1}{n} \sum_{i=1}^{n} y_i$. Therefore, based on formula (3.1), we can obtain that the coefficients of determination for the model fitting results in figure 4*a* and figure 4*b* are 0.92 and 0.9, respectively.

## 3.3. Sensitivity analysis

SA is a method to identify critical inputs (parameters and initial conditions) of a model and quantify how input uncertainty impacts model outcomes [38]. Among the alternative sampling-based methods for performing SA, partial rank correlation coefficient (PRCC), as a global technique, is one of the most efficient and reliable methods. Usually, the PRCC method is combined with the Latin hypercube sampling (LHS) method, which requires fewer samples than simple random sampling to achieve the same accuracy [39]. Therefore, we use the standard PRCC-LHS scheme to perform the SA, and then identify the key parameters contributing to the augmentation of the peak viral load and peak time in the primary infection and the secondary infection. Here, we explored the parameter space by performing an uncertainty analysis using the LHS method, and chose a normal distribution for all the input parameters with mean value and standard deviation in the absence of available data on the distribution functions. Furthermore, we use the *t*-test to perform the significance test to check if a PRCC is significantly different from zero.

## 4. Main results

Our estimations show that, owing to the pre-existing immunity to ZIKV, in the secondary DENV infection, the cross-immune response of ZIKV-specific antibody to DENV is built up faster and higher than the DENV-specific immune response with $A_z(0) > A_d(0)$ and $\kappa > a_d$. Similarly, in figure 5, it is shown that in the secondary DENV infection, the concentration of ZIKV-specific antibody is always higher than the DENV-specific antibody during the infectious period, which is in line with the experimental study [9]. Comparing the best fitting curves in figure 4*a* and *b*, we note that the peak viral load of the secondary infection is much higher than the peak viral load of the primary infection owing to the high level cross-immune response. Moreover, we observe that the peak of the primary infection is much more gradual than the peak of the secondary infection, while a sharp increase of the dengue viral load before the peak time and a sharp decrease after the peak time occurs (figure 4*b*), which is also consistent with the experimental observations in the study [9].

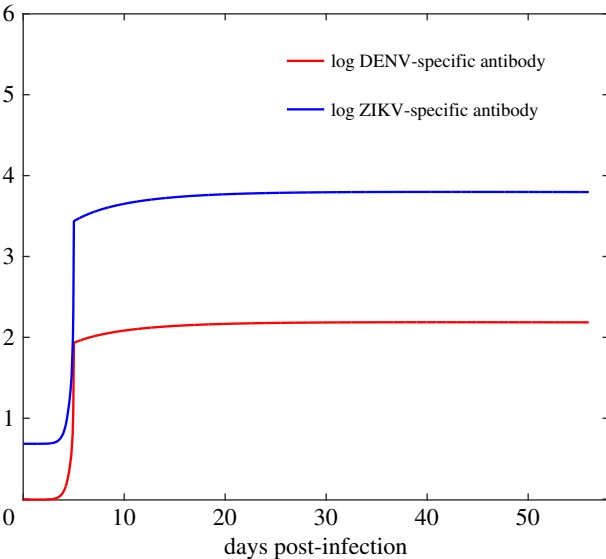

**Figure 5.** The curves of the concentrations of DENV-specific antibody and ZIKV-specific antibody by solving model (2.2). All the parameter values are fixed as the same as those in table 1.

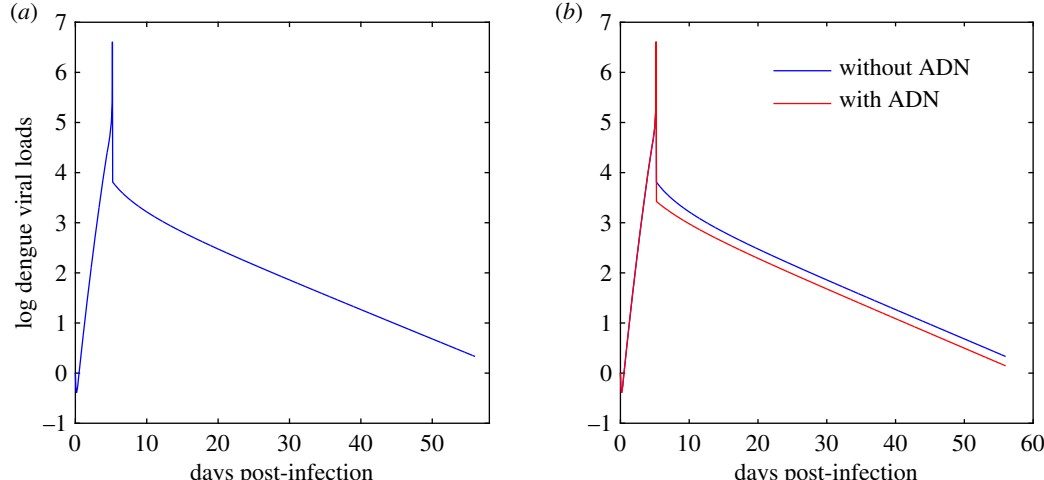

**Figure 6.** Solutions of model (2.2) with and without ADN. Red curve: the fitting curve of model (2.2). Blue curves: dengue viral loads by solving model (2.2) while we set $S(A_z) = 0$ when $A_z > A_{max}$ in contrast to the red curve. All the parameter values are fixed as the same as those in table 1.

In order to examine the key factors determining the sharp peak in the secondary infection, we set $S(A_z) = 0$ when $A_z > A_{max}$ (i.e. ignore the cross-neutralization of ZIKV-specific antibody to DENV) and plot the dengue viral loads in figure 6. It follows from figure 6 that ADN has little impact on the sharp peak viral load, while it can only help in clearing the DENV in the last stage of the infectious period. This indicates that ADE is the main factor contributing to the sharp increase and sharp decrease of the viral loads in the secondary infection.

In figure 7, we use a PRCC to conduct the SA of the peak dengue viral load and the peak time to all the estimated parameters. It follows from figure 7a that in the primary infection of DENV, there are four PRCC values that are significantly different from zero, including the infection rate $\beta_d$, the production rate of DENV-specific antibody $a_d$, the neutralization rate $b_d$, and the recruitment rate of target cell $\Lambda$. $\beta_d$ is ranked the first that is positively correlated to the peak dengue viral load among these examined parameters, while $\beta_d$ is the first parameter with negative impact on the peak time. This means that increasing the value of $\beta_d$ can greatly increase the peak dengue viral load with an earlier peak time. However, in the secondary DENV infection, the infection rate $\beta_d$ is of a very small PRCC value related to the peak virus load, as shown in figure 7b. Instead, $\theta_d$ and $A_{max}$ become the first two parameters

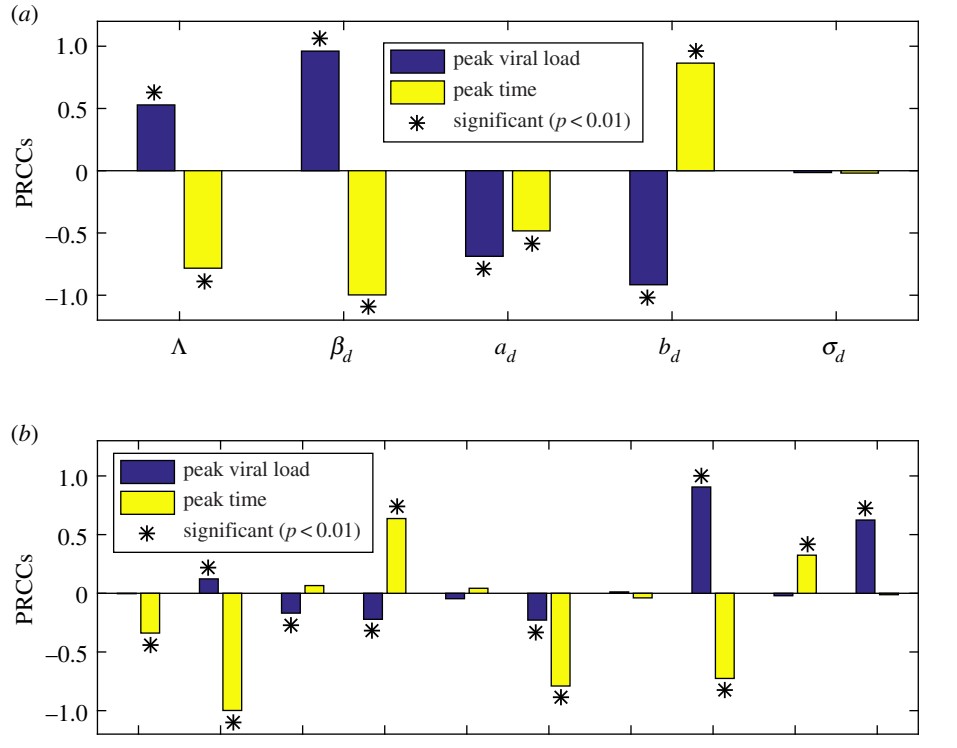

**Figure 7.** Sensitivity analysis. (*a*) PRCCs of the peak dengue viral load and the peak time for the primary DENV-infection (i.e. model (2.1)); (*b*) PRCCs of peak dengue viral load and the peak time for the secondary DENV-infection with a previous ZIKV-infection (i.e. model (2.2)). '*' denotes PRCCs that are significantly different from zero.

that are positively correlated to the peak viral load. This means that the high-level ADE effect and the delayed ADN effect of ZIKV-specific antibody to DENV are the key factors contributing to the much higher peak viral load in the secondary infection. We further noticed that the infection rate $\beta_d$ remains the first parameter highly correlated to the peak time in the secondary infection, i.e. the larger $\beta_d$, the earlier the peak time. According to figure 7*b*, the parameters $\theta_d$ and $A_{\max}$ have a similar impact on the peak time.

Next, we consider the impact of ADE on the virus dynamics of both DENV and ZIKV when an individual is coinfected with the two viruses. Two kinds of co-infection are considered here, i.e. simultaneous co-infection or subsequent co-infection. We initially assume that one individual is infected with DENV and ZIKV simultaneously. Based on the parameter estimations of model (2.1), we set the initial condition as $(1.98 \times 10^6, 0, 0, 1, 1, 0.988, 0.988)$ for model (2.4). Then, by changing the values of the parameters $\theta_d$ and $\theta_z$ and fixing all the other parameter values, we plot the dengue viral loads and the Zika viral loads in figure 8. Comparing figure 8 to figure 4*a*, we find that as for the simultaneous co-infection, the ADE effect has very limited impact on the virus dynamics of both DENV and ZIKV. That is, compared with the primary infection of a single virus, the ADE can lead to only a small increment of the peak viral load for the simultaneous co-infection. This indicates that the pre-existing immunity of one virus is the key determinant of the high-level ADE effect, consequently, resulting in the significant increase of the peak viral load of the other virus. This further indicates that for the subsequent co-infection, ADE may have a great influence on the virus dynamics of the subsequently-infected virus owing to the pre-existence of immunity to the initially-infected virus.

We further consider the impact of ADE on the virus dynamics of the subsequent co-infection. Without loss of generality, we assume that individuals are initially infected with ZIKV, and subsequently infected with DENV. That is, we assume that there is a time delay between the two infections, hereafter we call the interval of the delay as 'inter-infections interval (IFI)'. In figure 9, we set the IFI as 1 day and 5 days, respectively, and plot the dengue viral loads and Zika viral loads. We treat the initial ZIKV-infection as the primary infection, hence the Zika viral loads during the period of IFI (the grey curves in figure 9*a*,*c*) can be obtained by solving model (2.1) with the initial condition $(1.98 \times 10^6, 0, 1, 0.988)$. Then, at time $t = 0$ day one DENV is introduced with the initial values $I_d(0) = 0$

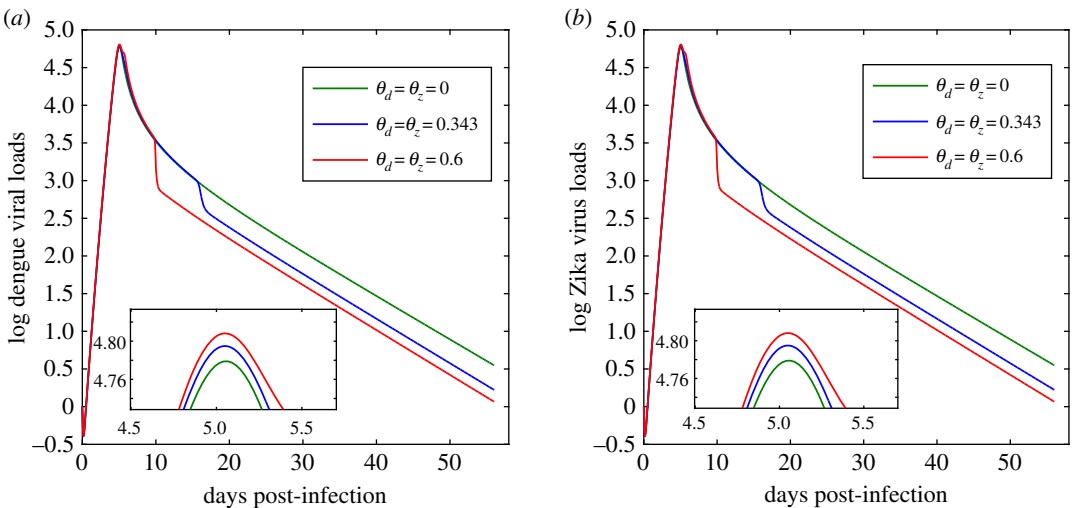

**Figure 8.** Solutions of model (2.4) in the case of simultaneous co-infection of DENV and ZIKV with different values of $\theta_d$ and $\theta_z$. (a) Dengue viral loads; (b) Zika viral loads. All the other parameter values are chosen from table 1. The small maps on the bottom are the partial enlarged drawing around the peak time.

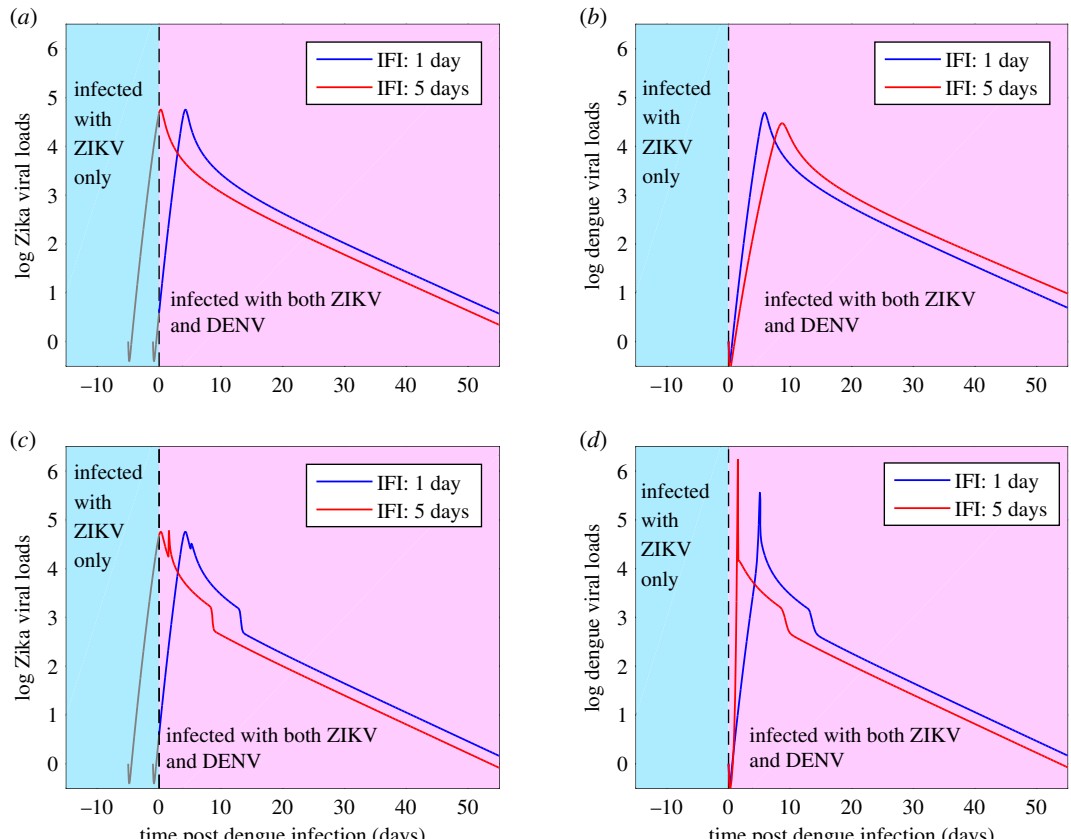

**Figure 9.** Dengue viral loads and Zika viral loads in the case of subsequent co-infection. Here, we set $\theta_d = \theta_z = 0$ in (a,b) and $\theta_d = \theta_z = 0.45$ in (c,d). The other parameter values are fixed as those in table 1.

and $A_d(0) = 0.988$. Therefore, for $t > 0$, the co-infection model (2.4) can be used to solve both the dengue viral loads and Zika viral loads (figure 9). Note that, in figure 9a and b, we set $\theta_d = \theta_z = 0$, that is, we ignore the cross-reactive response of the antibodies to the viruses. Consequently, the peak of the second virus (i.e. DENV) decreases (with a later peak time) as the IFI increases, as shown in figure 9b. By contrast, there is little influence on the initial virus (i.e. ZIKV) after the introduction of a second

virus in the absence of cross-reactive response between them, as shown in figure 9a. Under the same parameter setting of figure 9a,b, but including the cross-reactive response of the antibodies to the viruses by letting $\theta_d = \theta_z = 0.45$, figure 9c shows that the viral loads of ZIKV can have a second peak time owing to the ADE effect of DENV-specific antibody to ZIKV. Moreover, the ADE of the initial virus (ZIKV)-specific antibody to the second virus (DENV) can result in a very large peak viral load of the second virus with a much earlier peak time, as shown in figure 9d.

# 5. Discussion

In this study, we developed and examined a within-host mathematical model of secondary DENV infection with a primary infection of ZIKV, and a within-host mathematical model describing the virus dynamics for the co-infection of DENV and ZIKV. In our setting, we included the antibodies to describe the humoral immune response, and to parameterize the dynamical cross-immune response between DENV and ZIKV (i.e. the ADE and the ADN). This seems, to the best of our knowledge, the first attempt to provide a quantitative framework for understanding the impact of ADE and ADN effect between DENV and ZIKV on the virus dynamics of both viruses in general, and on the onset of severe disease by peak viral loads in particular.

By fitting the model to the data of dengue viral loads from the macaque infected with dengue only, we first calibrated the model parameters relevant to the primary dengue infection. Fixing the parameters corresponding to the primary dengue infection, we then fitted the model to the data of dengue viral loads from ZIKV convalescence macaque super-infected with DENV, and estimated the parameters related to cross-reactive response of ZIKV-specific antibody to DENV. From these estimated parameter values, we found that, owing to the pre-existence of immunity to ZIKV, the cross-immune response of ZIKV-specific antibody in the secondary DENV infection is much higher and is built-up much faster than the DENV-specific immune response, in line with the experimental study [9].

To model the impact of ADE, the existing studies [26,27] for modelling secondary DENV infection with a previous infection of another dengue serotype simply assumed that the humoral immune response stays at a constant level by re-parameterizing the infection rate to a larger value compared with the primary infection. This practice may oversimplify the contribution of antibodies to disease severity. By contrast, our models involved the antibodies to describe the humoral immune response, which can explicitly describe the dynamical antibody density-dependent cross-immune response between DENV and ZIKV. The data fitting results in figure 4 show that the primary DENV infection is characterized by a more gradual peak in viraemia while the secondary DENV infection has a very sharp peak in viraemia, also consistent with the study [9]. In other words, the secondary DENV infection with a primary ZIKV infection is of shorter duration than the primary dengue infection. In order to identify the key factors determining the sharp peak in the secondary infection, we plotted the viral loads in figure 6 by ignoring the ADN effect, and we concluded that ADN has very small impact on the sharp peak, including the sharp increase and the sharp decrease. This shows that ADE is the main factor determining the sharp increase/decrease of the viral loads around the peak time in the secondary infection.

To identify the key parameter of ADE contributing to the significant increment of the peak viral load in detail, we conducted an SA. The analysis shows that the infection rate $\beta_d$ is the most significant parameter with the largest PRCC value for the peak viral load in the primary infection while it has a very small PRCC value in the secondary infection. Instead, the parameters related to the cross-reactive response of ZIKV-specific antibody to DENV $\theta_d$, $A_{max}$ emerge as the parameters that are highly correlated to the peak viral load with great PRCC values. We, therefore, predict that the ADE of ZIKV to DENV can significantly increase the peak viral load in the secondary DENV infection. We also identified the infection rate $\beta_d$ as the most significant parameter negatively correlated to the peak time in both the primary infection and the secondary infection. Therefore, increasing the value of $\beta_d$ can significantly move the peak time ahead.

Through numerical simulations, we quantitatively examined the impact of ADE on the virus dynamics of both DENV and ZIKV when an individual is coinfected with the two viruses. In the current study, we considered two types of co-infection, i.e. simultaneous co-infection and subsequent co-infection. As for the case of simultaneous co-infection, we predict that the increment of the peak viral load induced by ADE is very small compared with sharp increase in the secondary infection. This means that the pre-existence of immunity to one virus is the determinant of the high-level ADE effect, which also indicates that the ADE effect may have a greater influence on the second virus during the subsequent co-infection.

Furthermore, we assumed that an individual was firstly infected with ZIKV and subsequently infected with DENV with a delay by a magnitude of several days, called the IFI between the two viruses. We found that in the absence of ADE between the two viruses, as the IFI increases, the peak viral load of the second virus decreases with a later peak time. Meanwhile, we found that there is little impact on the initial virus without ADE effect. This leads to the observation that subsequent co-infection is less damaging to the host compared with the simultaneous co-infection in the absence of ADE. However, if we include ADE, our simulations predict that the subsequent co-infection can result in a significant increment of the peak viral load of the second virus (DENV) with an earlier peak time. Moreover, owing to ADE, there can be a prolonged time period of the high level of viral loads or a second peak viral load for the initial virus (ZIKV). Therefore, we predict that owing to the ADE effect, the subsequent co-infection is more damaging to the host compared with the simultaneous co-infection.

It should be mentioned that the simplifying assumptions of our mathematical models result in a number of limitations. In the current study, we only included the immune response induced by antibodies. In reality, the immune response should be much more complex, and multiple immune cells, such as the T-cells and natural killer cells can play important roles in virus clearance [40]. We examined and predicted the impact of ADE on the viral dynamics of both DENV and ZIKV in the case of co-infection, however, the parameters of our co-infection model (i.e. model (2.4)) were not inferred from fitting experimental or clinical co-infection data. Instead, we used parameters inferred from fitting the data in the primary and secondary infection of DENV only. The study will benefit from future co-infection experiments. Despite these limitations, we believe that our study provides a solid qualitative framework for understanding the impact of the ADE effect between DENV and ZIKV on their virus dynamics, and hence, on the onset of severe disease.

In summary, we examined the impact of ADE and ADN effect between DENV and ZIKV on the onset of severe disease by peak viral load using mathematical models. We calibrated the model using the available data and our analyses are qualitatively in agreement with the experimental study, that the peak viral loads of the primary DENV infection is much more gradual compared with the secondary DENV infection with a primary ZIKV infection. Our numerical simulations predict that pre-existing immunity to ZIKV is the determinant of a high level of ADE effect to the secondary infection or the subsequent co-infection of DENV. Owing to the ADE effect, a subsequent co-infection can induce greater damage to the host with a higher peak viral load and a much earlier peak time for the second virus.

Data accessibility. The data used in this study are included in figure 3 accompanying with this manuscript. The package of MATLAB code for parameter estimation is available within the Dryad Digital Repository at: https://doi.org/10.5061/dryad.6q573n5vj [41].

Authors' contributions. B.T., Y.X., B.S. and J.W. designed the study and carried out the analysis. B.T. performed numerical simulations. All the authors contributed to writing the paper.

Competing interests. The authors declare that there is no conflict of interest regarding the publication of this paper.

Funding. This research is part of an international project entitled 'Research on Arbovirus Dynamics and Mitigation-Latin America and Canada' (RADAM-LAC), with field study sites in Colombia, Ecuador and Argentina. The RADAM-LAC Research Team consists of Beate Sander, Camila Gonzalez, Manisha Kulkarni, Marcos Miretti, Mauricio Espinel, Jianhong Wu and Varsovia Cevallos. This study was funded by a grant from the Canadian Institutes for Health Research (CIHR) and International Development Research Centre (IDRC)'s CIHR-IDRC Canada-Latin America and Caribbean Zika Virus Research Program to the RADAM-LAC Research Team.

Acknowledgements. We thank an associate editor and three reviewers for their constructive comments.

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
