## [Reviewer comments · Royal Society Open Science]

Review History

RSOS-191749.R0 (Original submission)

Review form: Reviewer 1

Is the manuscript scientifically sound in its present form?

Yes

Are the interpretations and conclusions justified by the results?

Yes

Is the language acceptable?

Yes

Do you have any ethical concerns with this paper?

No

Have you any concerns about statistical analyses in this paper?

No

Recommendation?

Reject

Comments to the Author(s)

Although the manuscript is mathematically sound. It is limited in scope and application. The authors have fit an ODE model to existing data but there are no clear predictions emanating from it that could be tested or verified. The text does not clearly state the conclusions or interpretation of the results.

Review form: Reviewer 2

Is the manuscript scientifically sound in its present form?

Yes

Are the interpretations and conclusions justified by the results?

Yes

Is the language acceptable?

Yes

Do you have any ethical concerns with this paper?

No

Have you any concerns about statistical analyses in this paper?

No

Recommendation?

Major revision is needed (please make suggestions in comments)

Comments to the Author(s)

The paper presents an interesting extension of infectious disease models to study co-infection. I appreciate the modifications the authors make to the topology of the network, and in particular, using data to calibrate the models. The main insight of the model is to draw attention to the antibody-dependent enhancement (ADE) as a determinant of the peak viral loads in the cases of secondary infection versus co-infection. However, I also have some concerns. I focus instead on the asymptotic persistence of viral loads in the data: Fig. 4 a and b clearly seem to indicate that viral loads persists at a steady level of 2 units even upto 50 days post-infection, whereas the model fits all show an almost linear decline in the viral loads. Is this model therefore a good model at all to work with? It did not sound convincing to me at all.

Review form: Reviewer 3

Is the manuscript scientifically sound in its present form?

No

Are the interpretations and conclusions justified by the results?

No

Is the language acceptable?

Yes

Do you have any ethical concerns with this paper?

No

Have you any concerns about statistical analyses in this paper?

Yes

Recommendation?

Major revision is needed (please make suggestions in comments)

Comments to the Author(s)

Antibody-dependent enhancement of dengue and related viruses is an important medical concern. Although, this phenomenon is well known, comprehensive mathematical models to understand it are scarce. Mathematical model presented in the current study is not the first attempt, since there are well known models on this aspect (see for example - Nikin-Beers & Ciupe, 2015, *Mathematical Biosciences*, 263, 83-92; Billings et al., 2007, *Journal of Theoretical Biology*, 246, 18-27), which authors have not referred to in the manuscript. This does not undermine the importance of model proposed in the current manuscript; however, authors should refer to these previous studies and compare the models and their performance. Model proposed in the current study is interesting and, although relatively simple, captures basic biology of interest. Authors' approach to fit their model to available data is also admirable. However, major concern regarding the manuscript is the lack of proper statistical treatment. Even though authors use terms such as "goodness of fit" (Figure 4 caption) and "significantly different from zero" (manuscript page 7, lines 18 to 20), authors have not performed any statistical test to support their claims. Detailed comments are provided below. Authors will need to revise the manuscript substantially before it can make significant contribution to science.

1. Statistical treatment

There are several issues with the statistical treatment that is essential and completely missing in the current manuscript.

(1) In Figure 2 authors provide some data from available literature. However, they do not mention what is the data provided. It is the average value that the bars depict? Also, what does the error bar refer to? Are these standard deviations or standard errors or confidence interval? In absence of this information the figure makes no sense.

(2) Further, in figure 4, authors provide the figure caption as "goodness of fit"; however, authors should understand that the term "goodness of fit" has a special meaning in statistical terminology, as it is a statistical estimate of how good the model fits to the data points. What authors show in the graph is model fit to the data points. Even then the graphs are incomplete because the observation points, which are as per Figure 2, are probably averages so there needs to be some estimate of uncertainty associated with them. In other terms, authors should provide the observation values with error bars and specify what the error bars refer to in the figure caption. Just revising the figure is also not sufficient, as authors have to prove that their model is a good fit to the data. Therefore, they actually need to estimate the statistical goodness of fit. There are several ways to do this, including log-likelihood based method, AUC curve and value, chi-square test, coefficient of determination, etc.

(3) Authors provide the analysis of partial rank correlation coefficients (PRCC) methods without properly explaining why and how it was performed in the methods section. Further, while explaining the results of Figure 7a, authors mention that four PRCC values were significantly different from zero; however, neither the figure nor the text provide any statistical inference to support the statement. If the figure authors should have provided 95% confidence interval of the estimates. When these intervals do not cross the zero, that is a good indication that the estimate is different from zero. For providing statistical significance to this, authors can perform one tailed t test to check the null hypothesis that the estimate is not different from zero and then use the p-

value corrected for family wise errors, because multiple tests are performed, to infer statistical significance.

2. Methodology

The methodology section is not in details and there are several issues, especially with respect to the statistical treatment.

- (1) While explaining the MCMC run, authors mention that the algorithm was run for 400,000 iterations with a burn-in of 400,000 iterations. In MCMC method, first few iterations are discarded as the estimates does not converge in the initial runs. This discard of the initial iterations is called burn-in. If authors are running 400,000 iterations and are discarding all 400,000 as burn-in, they will be left with no iterations for statistical estimates.
- (2) Partial rank correlation coefficients are introduced in the results section and there is no mention of this analysis in methods. Authors should provide details of why and how this analysis was performed in methods section.
- (3) There should be proper methodology defined for the goodness of fit of the model to the available data.

3. Manuscript preparation

Authors should proof read the manuscript properly.

- (1) There are spelling errors in several places. For example, "virous per day" (page 5, line 51-52) instead of "virus per day".
- (2) In panel figures authors should provide a general title to the figure before explaining the panels.
- (3) In figure 2, cite reference [9] for the data. Explain what are the bars (are these averages?) and what are the error bars (standard deviation, standard errors, or confidence intervals?).
- (4) Figure 4, "Goodness of fit" cannot be title of the figure because goodness of fit is a statistical estimate and cannot be used loosely. Provide error bars for the data points.
- (5) In figure 7 provide 95% CI for the estimates.
- (6) Authors should rename the "Conclusion and discussion" section as discussion section and provide final conclusion not more than a paragraph at the end of discussion or make a separate section for the same.

Decision letter (RSOS-191749.R0)

02-Jan-2020

Dear Dr Wu,

The editors assigned to your paper ("Modelling the impact of antibody-dependent enhancement on disease severity of ZIKV and DENV sequential and co-infection") have now received comments from reviewers. We would like you to revise your paper in accordance with the referee and Associate Editor suggestions which can be found below (not including confidential reports to the Editor). Please note this decision does not guarantee eventual acceptance.

Please submit a copy of your revised paper before 25-Jan-2020. Please note that the revision deadline will expire at 00.00am on this date. If we do not hear from you within this time then it will be assumed that the paper has been withdrawn. In exceptional circumstances, extensions may be possible if agreed with the Editorial Office in advance. We do not allow multiple rounds of revision so we urge you to make every effort to fully address all of the comments at this stage. If deemed necessary by the Editors, your manuscript will be sent back to one or more of the original reviewers for assessment. If the original reviewers are not available, we may invite new reviewers.

- Data accessibility

If you wish to submit your supporting data or code to Dryad (<http://datadryad.org/>), or modify your current submission to dryad, please use the following link:
<http://datadryad.org/submit?journalID=RSOS&manu=RSOS-191749>

- Competing interests

- Authors' contributions

AB carried out the molecular lab work, participated in data analysis, carried out sequence alignments, participated in the design of the study and drafted the manuscript; CD carried out the statistical analyses; EF collected field data; GH conceived of the study, designed the study,

coordinated the study and helped draft the manuscript. All authors gave final approval for publication.

- Acknowledgements

- Funding statement

on behalf of Dr Andrew Angel (Associate Editor) and Mark Chaplain (Subject Editor)
openscience@royalsociety.org

Associate Editor's comments (Dr Andrew Angel):

The reviewers believe overall that this research could provide a significant scientific contribution but that major changes are required before the manuscript is suitable for publication. One of the reviewers was in favour of rejection, in part on the grounds that the model doesn't provide any clear predictions - this is something that can be rectified in addition to responding to the suggestions of the other reviewers.

Associate Editor first Comments to the Author:

The manuscript present a study of the effects of antibody-dependent enhancement, ADE, (and to a lesser extent, antibody-dependent neutralization, ADN) on disease severity due to sequential and co-infection by the Dengue and Zika viruses. To the best of my knowledge, the study offers novel results and is scientifically sound. Therefore, I am recommending the manuscript for peer review.

Comments to Author:

Reviewers' Comments to Author:

Reviewer: 1

Comments to the Author(s)

Although the manuscript is mathematically sound. It is limited in scope and application. The authors have fit an ODE model to existing data but there are no clear predictions emanating from it that could be tested or verified. The text does not clearly state the conclusions or interpretation of the results.

Reviewer: 2

Comments to the Author(s)

The paper presents an interesting extension of infectious disease models to study co-infection. I appreciate the modifications the authors make to the topology of the network, and in particular, using data to calibrate the models. The main insight of the model is to draw attention to the

antibody-dependent enhancement (ADE) as a determinant of the peak viral loads in the cases of secondary infection versus co-infection. However, I also have some concerns. I focus instead on the asymptotic persistence of viral loads in the data: Fig. 4 a and b clearly seem to indicate that viral loads persists at a steady level of 2 units even upto 50 days post-infection, whereas the model fits all show an almost linear decline in the viral loads. Is this model therefore a good model at all to work with? It did not sound convincing to me at all.

Reviewer: 3

Comments to the Author(s)

Antibody-dependent enhancement of dengue and related viruses is an important medical concern. Although, this phenomenon is well known, comprehensive mathematical models to understand it are scarce. Mathematical model presented in the current study is not the first attempt, since there are well known models on this aspect (see for example - Nikin-Beers & Ciupe, 2015, *Mathematical Biosciences*, 263, 83-92; Billings et al., 2007, *Journal of Theoretical Biology*, 246, 18-27), which authors have not referred to in the manuscript. This does not undermine the importance of model proposed in the current manuscript; however, authors should refer to these previous studies and compare the models and their performance. Model proposed in the current study is interesting and, although relatively simple, captures basic biology of interest. Authors' approach to fit their model to available data is also admirable. However, major concern regarding the manuscript is the lack of proper statistical treatment. Even though authors use terms such as "goodness of fit" (Figure 4 caption) and "significantly different from zero" (manuscript page7, lines 18 to 20), authors have not performed any statistical test to support their claims. Detailed comments are provided below. Authors will need to revise the manuscript substantially before it can make significant contribution to science.

1. Statistical treatment

There are several issues with the statistical treatment that is essential and completely missing in the current manuscript.

(1) In Figure 2 authors provide some data from available literature. However, they do not mention what is the data provided. It is the average value that the bars depict? Also, what does the error bar refer to? Are these standard deviations or standard errors or confidence interval? In absence of this information the figure makes no sense.

(2) Further, in figure 4, authors provide the figure caption as "goodness of fit"; however, authors should understand that the term "goodness of fit" has a special meaning in statistical terminology, as it is a statistical estimate of how good the model fits to the data points. What authors show in the graph is model fit to the data points. Even then the graphs are incomplete because the observation points, which are as per Figure 2, are probably averages so there needs to be some estimate of uncertainty associated with them. In other terms, authors should provide the observation values with error bars and specify what the error bars refer to in the figure caption. Just revising the figure is also not sufficient, as authors have to prove that their model is a good fit to the data. Therefore, they actually need to estimate the statistical goodness of fit. There are several ways to do this, including log-likelihood based method, AUC cure and value, chi-square test, coefficient of determination, etc.

(3) Authors provide the analysis of partial rank correlation coefficients (PRCC) methods without properly explaining why and how it was performed in the methods section. Further, while explaining the results of Figure 7a, authors mention that four PRCC values were significantly different from zero; however, neither the figure nor the text provide any statistical inference to support the statement. If the figure authors should have provided 95% confidence interval of the estimates. When these intervals do not cross the zero, that is a good indication that the estimate is different from zero. For providing statistical significance to this, authors can perform one tailed t test to check the null hypothesis that the estimate is not different from zero and then use the p-value corrected for family wise errors, because multiple tests are performed, to infer statistical significance.

2. Methodology

The methodology section is not in details and there are several issues, especially with respect to the statistical treatment.

- (1) While explaining the MCMC run, authors mention that the algorithm was run for 400,000 iterations with a burn-in of 400,000 iterations. In MCMC method, first few iterations are discarded as the estimates does not converge in the initial runs. This discard of the initial iterations is called burn-in. If authors are running 400,000 iterations and are discarding all 400,000 as burn-in, they will be left with no iterations for statistical estimates.
- (2) Partial rank correlation coefficients are introduced in the results section and there is no mention of this analysis in methods. Authors should provide details of why and how this analysis was performed in methods section.
- (3) There should be proper methodology defined for the goodness of fit of the model to the available data.

3. Manuscript preparation

Authors should proof read the manuscript properly.

- (1) There are spelling errors in several places. For example, "virous per day" (page 5, line 51-52) instead of "virus per day".
- (2) In panel figures authors should provide a general title to the figure before explaining the panels.
- (3) In figure 2, cite reference [9] for the data. Explain what are the bars (are these averages?) and what are the error bars (standard deviation, standard errors, or confidence intervals?).
- (4) Figure 4, "Goodness of fit" cannot be title of the figure because goodness of fit is a statistical estimate and cannot be used loosely. Provide error bars for the data points.
- (5) In figure 7 provide 95% CI for the estimates.
- (6) Authors should rename the "Conclusion and discussion" section as discussion section and provide final conclusion not more than a paragraph at the end of discussion or make a separate section for the same.

Author's Response to Decision Letter for (RSOS-191749.R0)

See Appendix A.

RSOS-191749.R1 (Revision)

Review form: Reviewer 1

Is the manuscript scientifically sound in its present form?

Yes

Are the interpretations and conclusions justified by the results?

Yes

Is the language acceptable?

Yes

Do you have any ethical concerns with this paper?

No

Have you any concerns about statistical analyses in this paper?

No

Recommendation?

Accept with minor revision (please list in comments)

Comments to the Author(s)

The authors have greatly improved their manuscript and have given the statistical details that was missing. Therefore, although I rejected it in the first round, I agree for this be to published in Royal Society Open Science. I have only one concern, they are using parameters inferred from one virus infection model to two virus co-infection models. I am not sure whether this is valid and it needs some reassurance. The authors have to state this as a limitation of this study.

Review form: Reviewer 3

Is the manuscript scientifically sound in its present form?

Yes

Are the interpretations and conclusions justified by the results?

Yes

Is the language acceptable?

Yes

Do you have any ethical concerns with this paper?

No

Have you any concerns about statistical analyses in this paper?

No

Recommendation?

Accept as is

Comments to the Author(s)

Authors have revised the manuscript as per the suggestions on the earlier draft.

Decision letter (RSOS-191749.R1)

02-Mar-2020

Dear Dr Wu:

On behalf of the Editors, I am pleased to inform you that your Manuscript RSOS-191749.R1 entitled "Modelling the impact of antibody-dependent enhancement on disease severity of ZIKV and DENV sequential and co-infection" has been accepted for publication in Royal Society Open Science subject to minor revision in accordance with the referee suggestions. Please find the referees' comments at the end of this email.

The reviewers and Subject Editor have recommended publication, but also suggest some minor revisions to your manuscript. Therefore, I invite you to respond to the comments and revise your manuscript.

- Ethics statement

- Data accessibility

If you wish to submit your supporting data or code to Dryad (<http://datadryad.org/>), or modify your current submission to dryad, please use the following link:
<http://datadryad.org/submit?journalID=RSOS&manu=RSOS-191749.R1>

- Competing interests

- Authors' contributions

- Acknowledgements

- Funding statement

Please note that we cannot publish your manuscript without these end statements included. We have included a screenshot example of the end statements for reference. If you feel that a given

heading is not relevant to your paper, please nevertheless include the heading and explicitly state that it is not relevant to your work.

Because the schedule for publication is very tight, it is a condition of publication that you submit the revised version of your manuscript before 11-Mar-2020. Please note that the revision deadline will expire at 00.00am on this date. If you do not think you will be able to meet this date please let me know immediately.

on behalf of Dr Andrew Angel (Associate Editor) and Mark Chaplain (Subject Editor)

Associate Editor Comments to Author (Dr Andrew Angel):

Comments to the Author:

The revisions to the manuscript have been accepted by the reviewers. One reviewer has requested that a single statement of a potential limitation of the study be made. Therefore I am recommending that the manuscript be accepted once this change has been made.

Reviewer comments to Author:

Reviewer: 3

Comments to the Author(s)

Authors have revised the manuscript as per the suggestions on the earlier draft.

Reviewer: 1

Comments to the Author(s)

The authors have greatly improved their manuscript and have given the statistical details that was missing. Therefore, although I rejected it in the first round, I agree for this to be published in Royal Society Open Science. I have only one concern, they are using parameters inferred from one virus infection model to two virus co-infection models. I am not sure whether this is valid and it needs some reassurance. The authors have to state this as a limitation of this study.

Author's Response to Decision Letter for (RSOS-191749.R1)

See Appendix B.

Decision letter (RSOS-191749.R2)

12-Mar-2020

Dear Dr Wu,

It is a pleasure to accept your manuscript entitled "Modelling the impact of antibody-dependent enhancement on disease severity of ZIKV and DENV sequential and co-infection" in its current form for publication in Royal Society Open Science.

Due to rapid publication and an extremely tight schedule, if comments are not received, your paper may experience a delay in publication. Royal Society Open Science operates under a continuous publication model. Your article will be published straight into the next open issue and this will be the final version of the paper. As such, it can be cited immediately by other

researchers. As the issue version of your paper will be the only version to be published I would advise you to check your proofs thoroughly as changes cannot be made once the paper is published.

Kind regards,
Lianne Parkhouse
Editorial Coordinator
Royal Society Open Science
openscience@royalsociety.org

on behalf of Dr Andrew Angel (Associate Editor) and Mark Chaplain (Subject Editor)
openscience@royalsociety.org

Appendix A

Associate Editor's comments (Dr Andrew Angel):

The reviewers believe overall that this research could provide a significant scientific contribution but that major changes are required before the manuscript is suitable for publication. One of the reviewers was in favour of rejection, in part on the grounds that the model doesn't provide any clear predictions - this is something that can be rectified in addition to responding to the suggestions of the other reviewers.

Response: We appreciate the recognition of the novelty of our results. We admire the experience of the Associate Editor as indeed this lack of clear predictions can and has been rectified by addressing the comments from other reviewers. In fact, the first part of the simulations (Figs. 4 and 5) is dedicated to model validation and calibration, but the rest simulations all aim to make predictions (we sometimes used the word “show” or “demonstrate”, and sometimes we did not make the predictions more explicit. Most of these predictions are made for the considered case of simultaneous co-infection or subsequent co-infection of DENV and ZIKV (a primary ZIKV infection following by a secondary DENV infection several days later). We have rephrased the discussions appropriately, in addition to address some issues pointed out by the other two reviewers.

Associate Editor first Comments to the Author:

The manuscript present a study of the effects of antibody-dependent enhancement, ADE, (and to a lesser extent, antibody-dependent neutralization, ADN) on disease

severity due to sequential and co-infection by the Dengue and Zika viruses. To the best of my knowledge, the study offers novel results and is scientifically sound. Therefore, I am recommending the manuscript for peer review.

Response: We appreciate the positive recommendation.

Response to Reviewer #1

Comments: *Although the manuscript is mathematically sound. It is limited in scope and application. The authors have fit an ODE model to existing data but there are no clear predictions emanating from it that could be tested or verified. The text does not clearly state the conclusions or interpretation of the results.*

Response: The first part of the simulations (Figs. 4 and 5) is indeed dedicated to model validation and calibration. However, the rest of the simulations focus on using the parametrized model to conduct simulations which predict the feasible outcomes, in the considered case of simultaneous co-infection or subsequent co-infection of DENV and ZIKV (a primary ZIKV infection following by a secondary DENV infection several days later). We have rephrased the last discussion section appropriately to make some of the predictions more explicit. Some of the predictions have also been made clearer and more explicit after we addressed the comments from the other two reviewers, see discussions below.

Response to Reviewer #2

Comments: *The paper presents an interesting extension of infectious disease*

models to study co-infection. I appreciate the modifications the authors make to the topology of the network, and in particular, using data to calibrate the models. The main insight of the model is to draw attention to the antibody-dependent enhancement (ADE) as a determinant of the peak viral loads in the cases of secondary infection versus co-infection. However, I also have some concerns. I focus instead on the asymptotic persistence of viral loads in the data: Fig. 4 a and b clearly seem to indicate that viral loads persists at a steady level of 2 units even up to 50 days post-infection, whereas the model fits all show an almost linear decline in the viral loads. Is this model therefore a good model at all to work with? It did not sound convincing to me at all.

Response: In Fig. 4, we plotted the data and the fitting curves on \log_{10} -scale. Therefore, the viral loads are $10^{1.7}$, which are less than 100 copies, at the last data points. In contrast, the peak viral load can be $10^{6.5}$, which is more than 3,000,000 copies. Particularly, in the study (Best K, Guedj J, Madelain V, et al. 2017 *PNAS* (Ref [32])), the authors assumed that the viral load is undetectable when it is less than 200 copies. This means that the virus has been almost cleared 4 weeks post-infection. The fitting result that the viral loads will finally tend to zero are also in agreement with many studies on model fitting of dengue or Zika virus dynamics (see Clapham HE, Tricou V, et al. 2014, *J. R. Soc. Interface* (Ref [26]); Ben-Shachar R, Koelle K, 2015, *J. R. Soc. Interface* (Ref [27]); Best K, Guedj J, Madelain V, et al. 2017 *PNAS* (Ref [32]);).

Response to Reviewer #3

Comments: *Antibody-dependent enhancement of dengue and related viruses is*

an important medical concern. Although, this phenomenon is well known, comprehensive mathematical models to understand it are scarce. Mathematical model presented in the current study is not the first attempt, since there are well known models on this aspect (see for example - Nikin-Beers & Ciupe, 2015, Mathematical Biosciences, 263, 83-92; Billings et al., 2007, Journal of Theoretical Biology, 246, 18-27), which authors have not referred to in the manuscript. This does not undermine the importance of model proposed in the current manuscript; however, authors should refer to these previous studies and compare the models and their performance. Model proposed in the current study is interesting and, although relatively simple, captures basic biology of interest. Authors' approach to fit their model to available data is also admirable. However, major concern regarding the manuscript is the lack of proper statistical treatment. Even though authors use terms such as "goodness of fit" (Figure 4 caption) and "significantly different from zero" (manuscript page7, lines 18 to 20), authors have not performed any statistical test to support their claims. Detailed comments are provided below. Authors will need to revise the manuscript substantially before it can make significant contribution to science.

Response: Thanks for providing the information, we have added the references (Ref [23] and Ref [31] the reviewer mentioned, and also briefly commented on the significance of the scientific contribution of these studies. In addition, the original version already cited several modelling studies on ADE among different dengue serotypes in the population level (Refs [20-22]) or in the individual level (Refs [26-30]).

For the comments on the statistical treatment, please see the detail responses below.

Comments and responses:

1. Statistical treatment

There are several issues with the statistical treatment that is essential and completely missing in the current manuscript.

(1) In Figure 2 authors provide some data from available literature. However, they do not mention what is the data provided. It is the average value that the bars depict? Also, what does the error bar refer to? Are these standard deviations or standard errors or confidence interval? In absence of this information the figure makes no sense.

Response: We have added the information of the data in the caption of figure 2, indicating that the bars denote the mean values of the viral loads while the error bars represent their standard deviations.

(2) Further, in figure 4, authors provide the figure caption as “goodness of fit”; however, authors should understand that the term “goodness of fit” has a special meaning in statistical terminology, as it is a statistical estimate of how good the model fits to the data points. What authors show in the graph is model fit to the data points. Even then the graphs are incomplete because the observation points, which are as per Figure 2, are probably averages so there needs to be some estimate of uncertainty associated with them. In other terms, authors should provide the observation values with error bars and specify what the error bars refer to in the figure caption. Just

revising the figure is also not sufficient, as authors have to prove that their model is a good fit to the data. Therefore, they actually need to estimate the statistical goodness of fit. There are several ways to do this, including log-likelihood based method, AUC cure and value, chi-square test, coefficient of determination, etc.

Response: We have renamed the title of figure 4 “Goodness of fit” as “Results of model fitting”. The error bars (i.e. the standard deviations) of the observed data are given in figure 4 as well. We have also added one paragraph (the last paragraph) in the second subsection of the methods section (i.e. section 2.2) to show that we use the coefficient of determination (R^2) measure to estimate the goodness of fit for our model fitting results, and the values of R^2 are given in the end of this section.

(3) Authors provide the analysis of partial rank correlation coefficients (PRCC) methods without properly explaining why and how it was performed in the methods section. Further, while explaining the results of Figure 7a, authors mention that four PRCC values were significantly different from zero; however, neither the figure nor the text provide any statistical inference to support the statement. If the figure authors should have provided 95% confidence interval of the estimates. When these intervals do not cross the zero, that is a good indication that the estimate is different from zero. For providing statistical significance to this, authors can perform one tailed t test to check the null hypothesis that the estimate is not different from zero and then use the p-value corrected for family wise errors, because multiple tests are performed, to infer

statistical significance.

Response: We have added a subsection named “Sensitivity analysis” to explain why and how we use PRCC to perform sensitivity analysis. In this subsection, we also mentioned that we chose the t-test to perform the statistical significance test. Consequently, we calculated the p-values of Fig. 7 and marked the parameters (with $p < 0.01$) being significant different from zero using the marker “**”.

Comments and responses:

2. Methodology

The methodology section is not in details and there are several issues, especially with respect to the statistical treatment.

(1) While explaining the MCMC run, authors mention that the algorithm was run for 400,000 iterations with a burn-in of 400,000 iterations. In MCMC method, first few iterations are discarded as the estimates does not converge in the initial runs. This discard of the initial iterations is called burn-in. If authors are running 400,000 iterations and are discarding all 400,000 as burn-in, they will be left with no iterations for statistical estimates.

Response: There is a misunderstanding due to our presentation. Actually, we ran 800,000 iterations, of which the first 400,000 iterations were discarded as the burn-in while the last 400,000 iterations were used for statistical estimates. Hence, we have changed our presentation to “The algorithm is run for 800,000 iterations with a burn-

in of the first 400,000 iterations”.

(2) Partial rank correlation coefficients are introduced in the results section and there is no mention of this analysis in methods. Authors should provide details of why and how this analysis was performed in methods section.

Response: We have added a subsection named “Sensitivity analysis” in the methods section to give a detail explanation on why and how we use PRCC to perform the sensitivity analysis, where a well-known work (Ref [38]) on performing global uncertainty and sensitivity analysis was cited.

(3) There should be proper methodology defined for the goodness of fit of the model to the available data.

Response: In the second subsection of the methods section (i.e. section 2.2), we added one paragraph (the last paragraph) to show that we use the coefficient of determination (R^2) measure to estimate the goodness of fit for our model fitting results, and the values of R^2 are given in the section.

Comments and responses:

3. Manuscript preparation

Authors should proof read the manuscript properly.

(1) There are spelling errors in several places. For example, “virous per day” (page 5,

line 51-52) instead of "virus per day".

Response: Corrected. And we have carefully checked the whole manuscript in terms of the similar issues and made the revision correspondingly.

(2) In panel figures authors should provide a general title to the figure before explaining the panels.

Response: We have added a general title to all the panel figures.

(3) In figure 2, cite reference [9] for the data. Explain what are the bars (are these averages?) and what are the error bars (standard deviation, standard errors, or confidence intervals?).

Response: We have cited reference [9] in the caption of figure 2, and also added the explanation for the bars and the error bars.

(4) Figure 4, "Goodness of fit" cannot be title of the figure because goodness of fit is a statistical estimate and cannot be used loosely. Provide error bars for the data points.

Response: We have renamed the title of figure 2 "Goodness of fit" as "Results of model fitting". The error bars for the data are given in figure 4 as well.

(5) In figure 7 provide 95% CI for the estimates.

Response: As we chose the t-test for the statistical significance test, we have calculated the p-values and marked the parameters significantly different to zero in figure 7, instead of giving the 95% CI.

(6) Authors should rename the "Conclusion and discussion" section as discussion section and provide final conclusion not more than a paragraph at the end of discussion or make a separate section for the same.

Response: We have renamed the "Conclusion and discussion" section as "Discussion" section, and also added one paragraph (the last paragraph in the "Discussion" section) as a final conclusion of this study. This Discussion section also made some clarification about our predictions in the two cases we simulated.

Appendix B

Associate Editor Comments to Author (Dr Andrew Angel):

Comments to the Author:

The revisions to the manuscript have been accepted by the reviewers. One reviewer has requested that a single statement of a potential limitation of the study be made. Therefore I am recommending that the manuscript be accepted once this change has been made.

Response: We appreciate the positive recommendation. We have added the statement of the limitation, see the response to reviewer 1 for details.

Response to Reviewer #1

Comments: *The authors have greatly improved their manuscript and have given the statistical details that was missing. Therefore, although I rejected it in the first round, I agree for this be to published in Royal Society Open Science. I have only one concern, they are using parameters inferred from one virus infection model to two virus co-infection models. I am not sure whether this is valid and it needs some reassurance. The authors have to state this as a limitation of this study.*

Response: Thanks a lot for reconsidering our paper with the positive comment. We did previously mention that one limitation of our study is that the parameterizations of the co-infection model were not obtained by fitting real data (in the second last paragraph of our original version). Indeed this was not transparent enough, and we have revised this part accordingly. This revised statement is copied below:

We examined and predicted the impact of ADE on the viral dynamics of both DENV and ZIKV in the case of co-infection, however, the parameters of our co-infection model (i.e. model (4)) were not inferred from fitting experimental or clinical co-infection data. Instead, we used parameters inferred from fitting the data in the primary and secondary infection of DENV only. The study will benefit from future co-infection experiments.

Response to Reviewer #3

Comments: *Authors have revised the manuscript as per the suggestions on the earlier draft.*

Response: We appreciate the comment.